# VA-RED$^2$: Video Adaptive Redundancy Reduction

**Bowen Pan[1], Rameswar Panda[2], Camilo Fosco[1], Chung-Ching Lin[3], Alex Andonian[1],**
**Yue Meng[2], Kate Saenko[2,4], Aude Oliva[1,2], Rogerio Feris[2]**
[1]MIT CSAIL, [2]MIT-IBM Waston AI Lab, [3]Microsoft, [4]Boston University

## ABSTRACT

Performing inference on deep learning models for videos remains a challenge due to the large amount of computational resources required to achieve robust recognition. An inherent property of real-world videos is the high correlation of information across frames which can translate into redundancy in either temporal or spatial feature maps of the models, or both. The type of redundant features depends on the dynamics and type of events in the video: static videos have more temporal redundancy while videos focusing on objects tend to have more channel redundancy. Here we present a redundancy reduction framework, termed VA-RED$^2$, which is *input-dependent*. Specifically, our VA-RED$^2$ framework uses an input-dependent policy to decide how many features need to be computed for temporal and channel dimensions. To keep the capacity of the original model, after fully computing the necessary features, we reconstruct the remaining redundant features from those using cheap linear operations. We learn the adaptive policy jointly with the network weights in a differentiable way with a shared-weight mechanism, making it highly efficient. Extensive experiments on multiple video datasets and different visual tasks show that our framework achieves $20\% - 40\%$ reduction in computation (FLOPs) when compared to state-of-the-art methods without any performance loss. Project page: `http://people.csail.mit.edu/bpan/va-red/`.

## 1 INTRODUCTION

Large computationally expensive models based on 2D/3D convolutional neural networks (CNNs) are widely used in video understanding (Tran et al., 2015; Carreira & Zisserman, 2017; Tran et al., 2018). Thus, increasing computational efficiency is highly sought after (Feichtenhofer, 2020; Zhou et al., 2018c; Zolfaghari et al., 2018). However, most of these efficient approaches focus on architectural changes in order to maximize network capacity while maintaining a compact model (Zolfaghari et al., 2018; Feichtenhofer, 2020) or improving the way that the network consumes temporal information (Feichtenhofer et al., 2018; Korbar et al., 2019). Despite promising results, it is well known that CNNs perform unnecessary computations at some levels of the network (Han et al., 2015a; Howard et al., 2017; Sandler et al., 2018; Feichtenhofer, 2020; Pan et al., 2018), especially for video models since the high appearance similarity between consecutive frames results in a large amount of redundancy.

In this paper, we aim at dynamically reducing the internal computations of popular video CNN architectures. Our motivation comes from the existence of highly similar feature maps across both time and channel dimensions in video models. Furthermore, this internal redundancy varies depending on the input: for instance, static videos will have more temporal redundancy whereas videos depicting a single large object moving tend to produce a higher number of redundant feature maps. To reduce the varied redundancy across channel and temporal dimensions, we introduce an input-dependent redundancy reduction framework called VA-RED$^2$ (Video Adaptive REDundancy REDuction) for efficient video recognition (see Figure 1 for an illustrative example). Our method is model-agnostic and hence can be applied to any state-of-the-art video recognition networks.

The key mechanism that VA-RED$^2$ uses to increase efficiency is to replace full computations of some redundant feature maps with cheap reconstruction operations. Specifically, our framework avoids computing all the feature maps. Instead, we choose to only calculate those non-redundant part of feature maps and reconstruct the rest using cheap linear operations from the non-redundant

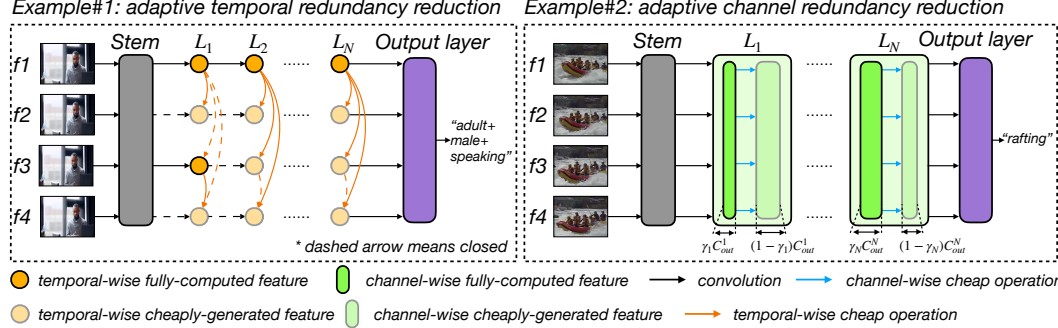

Figure 1: Our VA-RED[2] framework dynamically reduces the redundancy in two dimensions. Example 1 (left) shows a case where the input video has little movement. The features in the temporal dimension are highly redundant, so our framework fully computes a subset of features, and reconstructs the rest with cheap linear operations. In the second example, we show that our framework can reduce computational complexity by performing a similar operation over channels: only part of the features along the channel dimension are computed, and cheap operations are used to generate the rest.

features maps. In addition, VA-RED[2] makes decisions on a per-input basis: our framework learns an input-dependent policy that defines a "full computation ratio" for each layer of a 2D/3D network. This ratio determines the amount of features that will be fully computed at that layer, versus the features that will be reconstructed from the non-redundant feature maps. Importantly, we apply this strategy on both time and channel dimensions. We show that for both traditional video models such as I3D (Carreira & Zisserman, 2017), R(2+1)D (Tran et al., 2018), and more advanced models such as X3D (Feichtenhofer, 2020), this method significantly reduces the total floating point operations (FLOPs) on common video datasets without accuracy degradation.

The main **contributions** of our work includes: (1) A novel **input-dependent adaptive framework** for efficient video recognition, VA-RED[2], that automatically decides what feature maps to compute per input instance. Our approach is in contrast to most current video processing networks, where feature redundancy across both time and channel dimensions is not directly mitigated. (2) An **adaptive policy** jointly learned with the network weights in a fully differentiable way with a shared-weight mechanism, that allows us to make decisions on how many feature maps to compute. Our approach is model-agnostic and can be applied to any backbones to reduce feature redundancy in both time and channel domains. (3) **Striking results of VA-RED[2] over baselines**, with a 30% reduction in computation in comparison to R(2+1)D (Tran et al., 2018), a 40% over I3D-InceptionV2 (Carreira & Zisserman, 2017), and about 20% over the recently proposed X3D-M (Feichtenhofer, 2020) without any performance loss, for video action recognition task. The superiority of our approach is extensively tested on three video recognition datasets (Mini-Kinetics-200, Kinetics-400 (Carreira & Zisserman, 2017), and Moments-In-Time (Monfort et al., 2019)) and one spatio-temporal action localization dataset (J-HMDB-21 (Jhuang et al., 2013)). (4) A **generalization of our framework** to video action recognition, spatio-temporal localization, and semantic segmentation tasks, achieving promising results while offering significant reduction in computation over competing methods.

## 2 RELATED WORK

**Efficiency in Video Understanding Models.** Video understanding has made significant progress in recent years, mainly due to the adoption of convolutional neural networks, in form of 2D CNNs (Karpathy et al., 2014; Simonyan & Zisserman, 2014; Chéron et al., 2015; Feichtenhofer et al., 2017; Gkioxari & Malik, 2015; Wang et al., 2016; Zhou et al., 2018a; Lin et al., 2019; Fan et al., 2019) or 3D CNNs (Tran et al., 2015; Carreira & Zisserman, 2017; Hara et al., 2018; Tran et al., 2018). Despite promising results on common benchmarks, there is a significant interest in developing more efficient techniques and smaller models with reasonable performance. Previous works have shown reductions in computational complexity by using hybrid 2D-3D architectures (Xie et al., 2018; Zhou et al., 2018c; Zolfaghari et al., 2018), group convolution (Tran et al., 2019) or selecting salient clips (Korbar et al., 2019). Feichtenhofer et al., (Feichtenhofer et al., 2018) propose a dedicated low-framerate pathway. Expansion of 2D architectures through a stepwise expansion approach over the key variables such as temporal duration, frame rate, spatial resolution, network width, is

recently proposed in (Feichtenhofer, 2020). Diba et al. (Diba et al., 2019) learn motion dynamic of videos with a self-supervised task for video understanding. Fan et al. (Fan et al., 2020) incorporate a efficient learnable 3D-shift module into a 3D video network. Wang et al. (Wang et al., 2020) devise a correlation module to learn correlation along temporal dimension. Li et al. (Li et al., 2020) encode the clip-level ordered temporal information with a CIDC network. While these approaches bring considerable efficiency improvements, none of them dynamically calibrates the required feature map computations on a per-input basis. Our framework achieves substantial improvements in average efficiency by avoiding redundant feature map computation depending on the input.

**Adaptive Inference.** Many adaptive computation methods have been recently proposed with the goal of improving efficiency (Bengio et al., 2015; 2013; Veit & Belongie, 2018; Wang et al., 2018; Graves, 2016; Meng et al., 2021). Several works have been proposed that add decision branches to different layers of CNNs to learn whether to exit the network for faster inference (Yu et al., 2018; Figurnov et al., 2017; McGill & Perona, 2017; Teerapittayanon et al., 2016) Wang et al. (Wang et al., 2018) propose to skip convolutional blocks on a per input basis using reinforcement learning and supervised pre-training. Veit et al. (Veit & Belongie, 2018) propose a block skipping method controlled by samples from a Gumbel softmax, while Wu et al. (Wu et al., 2018) develop a reinforcement learning approach to achieve this goal. Adaptive computation time for recurrent neural networks is also presented in (Graves, 2016). SpotTune (Guo et al., 2019) learns to adaptively route information through finetuned or pre-trained layers. A few works have been recently proposed for selecting salient frames conditioned on the input (Yeung et al., 2016; Wu et al., 2019; Korbar et al., 2019; Gao et al., 2019) while recognizing actions in long untrimmed videos. Different from adaptive data sampling (Yeung et al., 2016; Wu et al., 2019; Korbar et al., 2019; Gao et al., 2019), in this paper, our goal is to remove feature map redundancy by deciding how many features need to be computed for temporal and channel dimensions per input basis, for efficient video recognition. AR-Net (Meng et al., 2020) recently learns to adaptively choose the resolution of input frames with several individual backbone networks for video inference. In contrast, our method focuses on reducing the redundancy in both temporal and channel dimension, and is applicable to both 3D and 2D models, while AR-Net is only for 2D model and it is focused on spatial resolution. Moreover, our method integrates all the inference routes into a single model which is in the almost same size to the original base model. Thus our model is significantly smaller than AR-Net in terms number of model parameters.

**Neural Architecture Search.** Our network learns the best internal redundancy reduction scheme, which is similar to previous work on automatically searching architectures (Elsken et al., 2018). Liu et al. (Liu et al., 2018) formulate the architecture search task in a differentiable manner; Cai et al. (Cai et al., 2018) directly learn architectures for a target task and hardware, Tan et al. (Tan & Le, 2019) design a compound scaling strategy that searches through several key dimensions for CNNs (depth, width, resolution). Finally, Tan et al. (Tan et al., 2019) incorporate latency to find efficient networks adapted for mobile use. In contrast, our approach learns a policy that chooses over full or reduced convolutions at inference time, effectively switching between various discovered subnetworks to minimize redundant computations and deliver high accuracy.

## 3 VIDEO ADAPTIVE REDUNDANCY REDUCTION

Our main goal is to automatically decide which feature maps to compute for each input video in order to classify it correctly with the minimum computation. The intuition behind our proposed method is that there are many similar feature maps along the temporal and channel dimensions. For each video instance, we estimate the ratio of feature maps that need to be fully computed along the temporal dimension and channel dimension. Then, for the other feature maps, we reconstruct them from those pre-computed feature maps using cheap linear operations.

**Approach Overview.** Without loss of generality, we start from a 3D convolutional network $\mathcal{G}$, and denote its $l^{th}$ 3D convolution layer as $f_l$, and the corresponding input and output feature maps as $X_l$ and $Y_l$ respectively. For each 3D convolution layer, we use a very lightweight policy layer $p_l$ denoted as *soft modulation gate* to decide the ratio of feature maps along the temporal and channel dimensions which need to be computed. As shown in Figure 2, for temporal-wise dynamic inference, we reduce the computation of 3D convolution layer by dynamically scaling the temporal stride of the 3D filter with a factor $R = 2^{p_l(X_l)[0]}$. Thus the shape of output $Y_l'$ becomes $C_{out} \times T_o/R \times H_o \times W_o$. To keep the same output shape, we reconstruct the remaining features based on $Y_l'$ as

$$Y_l[j + iR] = \begin{cases} \Phi_{i,j}^t(Y_l'[i]) & \text{if } j \in \{1, ..., R-1\} \\ Y_l'[i] & \text{if } j = 0 \end{cases}, i \in \{0, 1, ..., T_o/R - 1\}, \qquad (1)$$

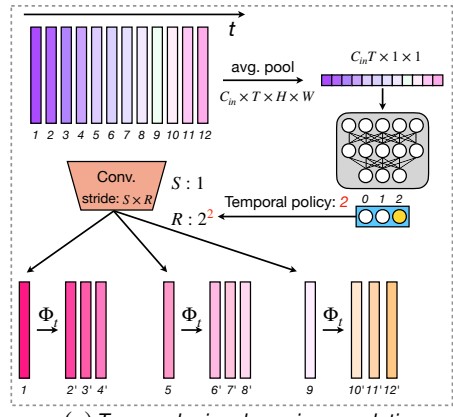 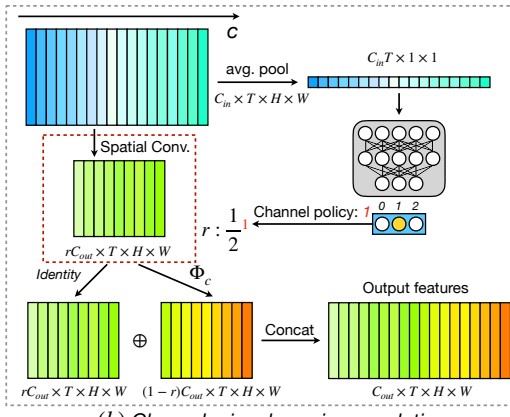

(a) Temporal-wise dynamic convolution      (b) Channel-wise dynamic convolution

Figure 2: An illustration of dynamic convolution along temporal dimension (a) and channel dimension (b) respectively. $\Phi_t$ and $\Phi_s$ represent the temporal cheap operation and spatial cheap operation respectively. In (a), we multiply the temporal stride $S$ with the factor $R = 2^{p_t}$ to reduce computation, where $p_t$ is the temporal policy output by soft modulation gate. In (b), we compute part of output features with the ratio of $r = (\frac{1}{2})^{p_c}$, where $p_c$ is the channel policy. Best viewed in color.

where $Y_l[j + iR]$ represents the $(j + iR)^{th}$ feature map of $Y_l$ along the temporal dimension, $Y'_l[i]$ denotes the $i^{th}$ feature map of $Y'_l$, and $\Phi^t_{i,j}$ is the cheap linear operation along the temporal dimension. The total computational cost of this process can be written as:

$$\mathcal{C}(f^t_l) = \frac{1}{R} \cdot \mathcal{C}(f_l) + \sum_{i,j} \mathcal{C}(\Phi^t_{i,j}) \approx \frac{1}{R} \cdot \mathcal{C}(f_l), \tag{2}$$

where the function $\mathcal{C}(\cdot)$ returns the computation cost for a specific operation, and $f^t_l$ represents our dynamic convolution process along temporal dimension. Different from temporal-wise dynamic inference, we reduce the channel-wise computation by dynamically controlling the number of output channels. We scale the output channel number with a factor $r = (\frac{1}{2})^{p_l(X_l)[1]}$. In this case, the shape of output $Y'_l$ is $rC_{out} \times T_o \times H_o \times W_o$. Same as before, we reconstruct the remaining features via cheap linear operations, which can be formulated as $Y_l = [Y'_l, \Phi^c(Y'_l)]$, where $\Phi^c(Y'_l) \in R^{(1-r)C_{out} \times T_o \times H_o \times W_o}$ represents the cheaply generated feature maps along the channel dimension, and $Y_l \in R^{C_{out} \times T_o \times H_o \times W_o}$ is the output of the channel-wise dynamic inference. The total computation cost of joint temporal-wise and channel-wise dynamic inference is:

$$\mathcal{C}(f^{t,c}_l) \approx \frac{r}{R} \cdot \mathcal{C}(f_l), \tag{3}$$

where $f^{t,c}_l$ is the adjunct process of temporal-wise and channel-wise dynamic inference.

**Soft Modulation Gate for Differentiable Optimization.** We adopt an extremely lightweight policy layer $p_l$ called soft modulation gate for each convolution layer $f_l$ to modulate the ratio of features which need to be computed. Specifically, the soft modulation gate takes the input feature maps $X_l$ as input and learns two probability vectors $V^l_t \in R^{S_t}$ and $V^l_c \in R^{S_c}$, where $S_t$ and $S_c$ are the temporal search space size and the channel search space size respectively. The $V^l_t$ and $V^l_c$ are learned by:

$$[V^l_t, V^l_c] = p_l(X_l) = \phi(\mathcal{F}(\omega_{p,2}, \delta(\mathcal{N}(\mathcal{F}(\omega_{p,1}, G(X_l)))))) + \beta^l_p), \tag{4}$$

where $\mathcal{F}(\cdot, \cdot)$ denotes the fully-connected layer, $\mathcal{N}$ is the batch normalization, $\delta(\cdot)$ represents the $\tanh(\cdot)$ function, $G$ is the global pooling operation whose output shape is $C_{in} \cdot T \times 1 \times 1$, $\phi(\cdot)$ is the output activation function, here we just use $\max(\tanh(\cdot), 0)$ whose output range is $[0, 1)$, and $\omega_{p,1} \in R^{(S_t + S_c) \times D_h}$, $\omega_{p,2} \in R^{D_h \times C_{in} \cdot T}$ are the weights of their corresponding layers, $D_h$ is the hidden dimension number. $V^l_t$ and $V^l_c$ will then be used to modulate the ratio of the feature maps to be computed in temporal dynamic convolution and channel-wise dynamic convolution. During training, we obtain the final output of the dynamic convolution by weighted sum of all the feature

maps which contains different ratio of fully-computed features as follows:

$$Y_c^l = \sum_{i=1}^{S_c} V_c^l[i] \cdot f_l^c(X_l, r = (\frac{1}{2})^{(i-1)}), \quad Y_l = \sum_{j=1}^{S_t} V_t^l[j] \cdot f_l^t(Y_c^l, R = 2^{(j-1)}), \quad (5)$$

where $f_l^c(\cdot, r)$ is the channel-wise dynamic convolution with the channel scaling factor $r$, and $f_l^t(\cdot, R)$ it the temporal-wise dynamic convolution with the temporal stride scaling factor $R$. During the inference phase, only the dynamic convolutions whose weights are not zero will be computed.

**Shared-weight Training and Inference.** Many works in adaptive computation and neural architecture search suffer from very heavy computational cost and memory usage during training stage due to the large search space. In our case, under the naive implementation, the training computational cost and parameter size would linearly grow as the search space size increases. To train our model efficiently, we utilize a weight-sharing mechanism to reduce the computational cost and training memory. To be specific, we first compute all the possible necessary features using a big kernel. Then, for each dynamic convolution with different scaling factor, we sample its corresponding ratio of necessary features and reconstruct the rest features by cheap operations to get the final output. Though this, we are able to keep the computational cost at a constant value invariant to the search space. More details on this are included in Section B of the Appendix.

**Efficiency Loss.** To encourage our network to output a computational efficient subgraph, we introduce the efficiency loss $\mathcal{L}_c$ during the training process, which can be formulated as

$$\mathcal{L}_e = (\mu_0 \sum_{l=1}^{L} \frac{\mathcal{C}(f_l)}{\sum_{k=1}^{L} \mathcal{C}(f_k)} \cdot \frac{r_l^s}{R_l^s})^2, \mu_0 = \left\{ \begin{array}{ll} 1 & \text{if correct} \\ 0 & \text{otherwise} \end{array} \right. , \quad (6)$$

where $r_l^s$ is channel scaling factor of the largest filter in the series of channel-wise dynamic convolutions, and $R_l^s$ is stride scaling factor of the largest filter of temporal-wise dynamic convolutions. Overall, the loss function of our whole framework can be written as $\mathcal{L} = \mathcal{L}_a + \lambda_e \mathcal{L}_e$, where $L_a$ is the accuracy loss of the whole network and $\lambda_e$ is the weight of efficiency loss which can be used to balance the importance of the optimization of prediction accuracy and computational cost.

## 4 EXPERIMENTS

**Datasets.** We conduct our **video action recognition** experiments on three standard benchmarks: Mini-Kinetics-200, Kinetics-400, and Moments-In-Time. Mini-Kinetics-200 (assembled by (Meng et al., 2020)) is a subset of full Kinetics dataset (Carreira & Zisserman, 2017) containing 121k videos for training and 10k videos for testing across 200 action classes. Moments-In-Time dataset has 802,244 videos in training and 33,900 videos in validation across 339 categories. To show the generalization ability to different task, we also conduct the **video spatio-temporal action localization** on J-HMDB-21 (Jhuang et al., 2013). J-HMDB-21 is a subset of HMDB dataset (Kuehne et al., 2011) which has 928 short videos with 21 action categories. We report results on the first split. For **semantic segmentation** experiments, we use ADE20K dataset (Zhou et al., 2017; 2018b), containing 20k images for training and 2k images for validation. ADE20K is a densely labeled image dataset where objects and object parts are segmented down to pixel level. We report results on validation set.

**Model Architectures.** We evaluate our method on three most widely-used model architectures: I3D (Carreira & Zisserman, 2017), R(2+1)D (Tran et al., 2018), and the recent efficient model X3D (Feichtenhofer, 2020). We consider I3D-InceptionV2 (denoted as I3D below) and R(2+1)D-18 (denoted as R(2+1)D below) as our base model. In our implementation of X3D, we remove all the swish non-linearity (Ramachandran et al., 2017) except those in SE layer (Hu et al., 2018) to save training memory and speed up the inference speed on GPU. We choose X3D-M (denote as X3D below) as our base model and demonstrate that our method is generally effective across datasets.

**Implementation Details.** We train and evaluate our baseline models by mainly following the settings in their original papers (Tran et al., 2018; Xie et al., 2018; Feichtenhofer, 2020). We train all our base and dynamic models for 120 epochs on mini-Kinetics-200, Kinetics-400, and 60 epochs on Moments-In-Time dataset. We use a mini-batch size of 12 clips per GPU and adopt synchronized SGD with cosine learning rate decaying strategy (Loshchilov & Hutter, 2016) to train all our models. Dynamic models are finetuned with efficiency loss for 40/20 epochs to reduce density of inference

Table 1: **Action recognition results using different number of input frames and different search space.** We choose R(2+1)D-18 on Mini-Kinetics-200 and study the performance with different number of input frames and different search space (denoted as Sea. Sp.). Search space of 2 means that both temporal-wise and channel-wise policy network have 2 alternatives: computing all feature maps, or computing only $\frac{1}{2}$) of the feature maps. Similarly, search space 3 have 3 alternatives: computing *1)* all feature maps, *2)* $\frac{1}{2}$ of feature maps, *3)* $\frac{1}{4}$ of feature maps. ✗ denote the base model and ✓ denote the dynamic model trained using our proposed approach VA-RED$^2$. We also report the average speed of different models in terms of number of clips processed in one second ($clip/second$).

| length | sp. | GFLOPs$_{Avg}$ | GFLOPs$_{Max}$ | GFLOPs$_{Min}$ | avg speed | clip-1 | video-1 | video-5 |
|---|---|---|---|---|---|---|---|---|
| | ✗ | 27.7 | 27.7 | 27.7 | 192.1 | 56.4 | 66.8 | 86.8 |
| 8 | 2 | 20.0(−28%) | 22.1(−20%) | 18.0(−35%) | **205.5** | 57.7 | **68.0** | **87.4** |
| | 3 | 21.6(−22%) | 23.2(−16%) | 19.8(−29%) | 201.4 | **58.2** | 67.7 | **87.4** |
| 16 | ✗ | 55.2 | 55.2 | 55.2 | 97.1 | 57.5 | 67.5 | 87.1 |
| | 2 | 40.4(−27%) | 43.2(−22%) | 36.6(−34%) | **108.7** | **60.6** | **70.0** | **88.7** |
| 32 | ✗ | 110.5 | 110.5 | 110.5 | 49.6 | 60.5 | 69.4 | 88.2 |
| | 2 | 79.3(−28%) | 89.5(−19%) | 72.4(−34%) | **53.4** | **63.3** | **72.3** | **89.7** |

Table 2: **Action recognition results on Mini-Kinetics-200.** We set the search space as 2 and train all the models with 16 frames. The metric speed uses $clip/second$ as the unit.

| Model | Dy. | GFLOPs | Speed | clip-1 | video-1 |
|---|---|---|---|---|---|
| R(2+1)D | ✗ | 55.2 | 97.1 | 57.5 | 67.5 |
| | ✓ | 40.4 | **108.7** | **60.6** | **70.0** |
| I3D | ✗ | 56.0 | 116.4 | 59.7 | 68.3 |
| | ✓ | 26.5 | **141.7** | **62.2** | **71.1** |
| X3D | ✗ | 6.20 | 169.4 | **66.5** | **72.2** |
| | ✓ | 5.03 | **178.2** | 65.5 | 72.1 |

Table 3: **Action recognition results with Temporal Pyramid Network (TPN) on Mini-Kinetics-200.** TPN-8f and TPN-16f indicate that we use 8 frames and 16 frames as input to the model respectively.

| Model | Dy. | GFLOPs | clip-1 | video-1 |
|---|---|---|---|---|
| TPN-8f | ✗ | 28.5 | 58.9 | 67.2 |
| | ✓ | 21.5 | **59.2** | **68.8** |
| TPN-16f | ✗ | 56.8 | 59.8 | 68.5 |
| | ✓ | 41.5 | **60.8** | **70.6** |

graph while maintaining the accuracy. During finetuning, we set $\lambda_c$ to 0.8 and learning rate to 0.01 for R(2+1)D and 0.1 for I3D and X3D. For testing, we adopt *K-LeftCenterRight* strategy: $K$ temporal clips are uniformly sampled from the whole video, on which we sample the left, center and right crops along the longer spatial axis, the final prediction is obtained by averaging these $3 \times K$ clip predictions. We set $K = 10$ on Mini-Kinetics-200 and Kinetics-400 and $K = 3$ on Moments-In-Time. More implementation details are included in Section B of Appendix. For video spatio-temporal action localization, we adopt YOWO architecture in (Köpüklü et al., 2019) and replace 2D branch with 3D backbone to directly compare them. We freeze the parameters of 3D backbone as suggested in (Köpüklü et al., 2019) due to small number of training video in J-HMDB-21 (Jhuang et al., 2013). The rest part of the network is optimized by SGD with initial learning rate of $10^{-4}$. Learning rate is reduced with a decaying factor of 0.5 at 10k, 20k, 30k and 40k iterations. For semantic segmentation, we conduct experiments using PSPNet (Zhao et al., 2017), with dilated ResNet-18 (Yu et al., 2017; He et al., 2016) as our backbone architecture. As PSPNet is devised for image semantic segmentation, we only apply the channel-wise redundancy reduction to the model and adopt synchronized SGD training for 100k iterations across 4GPUs with 2 images on each GPU. The learning rate decay follows the cosine learning rate schedule schedule (Loshchilov & Hutter, 2016).

**Results on Video Action Recognition.** We first evaluate our method by applying it to R(2+1)D-18 (Tran et al., 2018) with different number of input frames and different size of search space. Here we use GFLOPs (floating point operations) to measure the computational cost of the model and report clip-1, video-1 and video-5 metrics to measure the accuracy of our models, where clip-1 is the top-1 accuracy of model evaluation with only one clip sampled from video, video-1 and video-5 are the top-1 and top-5 accuracy of model evaluated with *K-LeftCenterRight* strategy. Note that we report the FLOPs of a single video clips at the spatial resolution $256 \times 256$ (for I3D and X3D) or $128 \times 128$ (for R(2+1)D). In addition, we report the speed of each model with the metric of $clip/second$, which denotes the number of video clips that are processed in one second. We create the environment with PyTorch 1.6, CUDA 11.0, and a single NVIDIA TITAN RTX (24GB) GPU as our testbed to measure speed of different models. Table 1 shows the results (In all of the tables, ✗ represents the

Table 4: **Comparison with CorrNet (Wang et al., 2020) and AR-Net (Meng et al., 2020) on Mini-Kinetics-200.** We set the search space as 2 and train all the models with 16 frames.

| Model | Dy. | GFLOPS | clip-1 | video-1 | Method | Params | GFLOPs | clip-1 |
|---|---|---|---|---|---|---|---|---|
| CorrNet | ✗ | 60.8 | 59.9 | 68.2 | AR-Net | 63.0M | 44.8 | 67.2 |
| | ✓ | **45.5** | **60.4** | **70.0** | VA-RED$^2$ | **23.9M** | **43.4** | **68.3** |

Table 5: **Action recognition results on Kinetics-400.** We set the search space as 2, meaning models can choose to compute all feature maps or $\frac{1}{2}$ of them both on temporal and channel-wise convolutions.

| Model | Dy. | 16-frame | | | | | 32-frame | | | | |
|---|---|---|---|---|---|---|---|---|---|---|---|
| | | GFLOPs | speed | clip-1 | video-1 | video-5 | GFLOPs | speed | clip-1 | video-1 | video-5 |
| R(2+1)D | ✗ | 55.2 | 97.1 | 57.3 | 65.6 | 86.3 | 110.5 | 49.6 | 61.5 | 69.0 | 88.6 |
| | ✓ | 40.3 | **105.9** | **58.4** | **67.6** | **87.6** | 80.7 | **53.0** | 61.5 | **70.0** | **88.9** |
| I3D | ✗ | 56.0 | 116.4 | 55.1 | 66.5 | 86.7 | 112.0 | 57.6 | 57.2 | 64.9 | 86.5 |
| | ✓ | 32.1 | **140.7** | **58.6** | **67.1** | **87.2** | 64.3 | **71.7** | **61.0** | **68.6** | **88.4** |
| X3D | ✗ | 6.42 | 169.4 | 63.2 | 70.6 | 90.0 | [X3D-M is designed for 16 frames] | | | | |
| | ✗ | 5.38 | **177.6** | **65.3** | **72.4** | **90.7** | | | | | |

original fixed model architecture while ✓ denote the dynamic model trained using our proposed approach). Our proposed approach VA-RED$^2$ significantly reduces the computational cost while improving the accuracy. We observe that dynamic model with the search space size of 2 has the best performance in terms of accuracy, GFLOPS and speed. We further test our VA-RED$^2$ with all of the three model architectures: R(2+1)D-18, I3D-InceptionV2, and X3D-M (Table 2) including the very recent temporal pyramid module (Yang et al., 2020) and correlation module (Wang et al., 2020) on Mini-Kinetics-200 dataset. We choose R(2+1)D-18 with TPN and CorrNet as the backbone architecture and test the performance of our method using a search space of 2 in Table 3 and Table 4 (Left) respectively. Table 2 shows that method boosts the speed of base I3D-InceptionV2 and R(2+1)D models by 21.7% and 10.6% respectively, showing its advantages not only in terms of GFLOPS but also in actual speed. Table 4 (Left) shows that our dynamic approach also outperforms the baseline CorrNet by 1.8% in top-1 video accuracy, while reducing the computational cost by 25.2% on Mini-Kinetics-200. Furthermore, we compare our method with AR-Net (Meng et al., 2020), which is a recent adaptive method that selects optimal input resolutions for video inference. We conduct our experiments on 16-frame TSN (Wang et al., 2016) with ResNet50 backbone and provide the comparison on FLOPs, parameter size, and accuracy (Table 4 (Right)). To make a fair comparison, we train AR-Net using the official implementation on the same Mini-Kinetics-200 dataset with Kaiming initialization (He et al., 2015). Table 4 (Right) shows that our method, VA-RED$^2$ outperforms AR-Net in both accuracy and GFLOPS, while using about 62% less parameters. Table 5 and Table 6 show the results of different methods on Kinetics-400 and Moments-In-Time, respectively. To summarize, we observe that VA-RED$^2$ consistently improves the performance of all the base models including the recent architectures X3D, TPN, and CorrNet, while offering significant reduction in computation. Moreover, our approach is model-agnostic, which allows this to be served as a plugin operation for a wide range of action recognition architectures. From the comparison among different models, we find that our proposed VA-RED$^2$ achieves the most computation reduction on I3D-InceptionV2, between 40% and 50%, while reducing less than 20% on X3D-M. This is because X3D-M is already very efficient both in terms of channel dimension and temporal dimension. Notice that the frames input to X3D-M are at the temporal stride of 5, which makes them share less similarity. Furthermore, we observe that dynamic I3D-InceptionV2 has very little variation of the computation for different input instances. This could be because of the topology configuration of the InceptionV2, which has lots of parallel structures inside the network architecture.

We also compare VA-RED$^2$ with a weight-level pruning method (Han et al., 2015b) and a automatic channel pruning method (CGNet) (Hua et al., 2019) on Mini-Kinetics-200. Table 7 shows that our approach significantly outperforms the weight-level pruning method by a margin of about 3%-4% in clip-1 accuracy with similar computation over the original fixed model and consistently outperforms CGNet while requiring less GFLOPs (maximum 2.8% in 16 frame). These results well demonstrate the effectiveness of our dynamic video redundancy framework over network pruning methods.

Table 6: **Action recognition results on Moments-In-Time.** We set the search space as 2, i.e., models can choose to compute all feature maps or $\frac{1}{2}$ of them both on temporal and channel-wise convolutions. The speed uses $clip/second$ as the unit.

| Model | Dy. | GFLOPs | speed | clip-1 | video-1 |
|-------|-----|--------|-------|--------|---------|
| R(2+1)D | ✗ | 55.2 | 97.1 | 27.0 | 28.8 |
|         | ✓ | 42.5 | **105.5** | **27.3** | **30.1** |
| I3D | ✗ | 56.0 | 116.4 | 25.7 | 26.8 |
|     | ✓ | 32.1 | **140.7** | **26.3** | **28.5** |
| X3D | ✗ | 6.20 | 169.4 | 24.8 | 24.8 |
|     | ✓ | 5.21 | **177.4** | **26.7** | **27.7** |

Table 7: **Comparison with network pruning methods.** We choose R(2+1)D on Mini-Kinetics-200 dataset with different number of input frames. Numbers in green/blue quantitatively show how much our proposed method is better/worse than these pruning methods.

| Method | Frames | GFLOPs | clip-1 |
|--------|--------|--------|--------|
| Weight-level | 8 | 19.9 (+3.2) | 54.5 (+3.2) |
|              | 16 | 40.3 (-0.1) | 57.7 (+2.9) |
|              | 32 | 79.6 (-0.3) | 59.6 (+3.7) |
| CGNet | 8 | 23.8 (+3.8) | 56.2 (+1.5) |
|       | 16 | 47.6 (+7.2) | 57.8 (+2.8) |
|       | 32 | 95.3 (+16.0) | 61.8 (+1.5) |

Table 8: **Action localization results on J-HMDB.** We set the search space as 2 for dynamic models. The speed uses $clip/second$ as the unit.

| Model | Dy. | GFLOPs | speed | mAP | Recall | Classif. |
|-------|-----|--------|-------|-----|--------|----------|
| I3D | ✗ | 43.9 | 141.1 | 44.8 | **67.3** | 87.2 |
|     | ✓ | 21.3 | **167.4** | **47.2** | 65.6 | **91.1** |
| X3D | ✗ | 5.75 | 176.3 | 47.9 | 65.2 | **93.2** |
|     | ✓ | 4.85 | **184.6** | **50.0** | **65.8** | 93.0 |

Table 9: **Effect of efficiency loss on Kinetics-400.** *Eff.* denotes the efficiency loss.

| Model | *Eff.* | GFLOPs | clip-1 | video-1 |
|-------|--------|--------|--------|---------|
| R(2+1)D | No | 49.8 | 57.9 | 66.7 |
|         | Yes | 40.3 | **58.4** | **67.6** |
| I3D | No | 56.0 | 58.0 | 66.5 |
|     | Yes | 32.1 | **58.6** | **67.1** |

Table 10: **Ablation experiments on dynamic modeling along temporal and channel dimensions.** We choose R(2+1)D-18 on Mini-Kinetics-200 and set the search space to 2 in all the dynamic models.

| Dy. Temp. | Dy. Chan. | 8-frame | | | | 16-frame | | | |
|-----------|-----------|---------|-------|--------|---------|----------|-------|--------|---------|
|           |           | GFLOPs | speed | clip-1 | video-1 | GFLOPs | speed | clip-1 | video-1 |
| ✗ | ✗ | 27.7 | 192.1 | 56.4 | 66.8 | 55.2 | 97.1 | 57.5 | 67.5 |
| ✓ | ✗ | 23.5 | 198.6 | 57.1 | 66.8 | 46.1 | 105.0 | 58.6 | 67.6 |
| ✗ | ✓ | 22.7 | 196.5 | 57.0 | 66.7 | 46.3 | 102.0 | 59.2 | 68.3 |
| ✓ | ✓ | 20.0 | **205.5** | **57.7** | **68.0** | 40.4 | **108.7** | **60.6** | **70.0** |

**Results on Spatio-Temporal Action Localization.** We further extend our method to the spatio-temporal action localization task to demonstrate the generalization ability to different task. We conduct our method on J-HMDB-21 with two different 3D backbone networks: I3D-InceptionV2 and X3D-M. We report frame-mAP at IOU threshold 0.5, recall value at IOU threshold 0.5, and classification accuracy of correctly localized detections to measure the performance of the detector. Table 8 shows that our dynamic approach outperforms the baselines on all three metrics while offering significant savings in FLOPs (e.g., more than 50% savings on I3D). In summary, VA-RED[2] is clearly better than the baseline architectures in terms of both accuracy and computation cost on both recognition and localization tasks, making it suitable for efficient video understanding.

**Effect of Efficiency Loss.** We conduct an experiment by comparing the model performance before and after being finetuned with our proposed efficiency loss. Table 9 shows that finetuning our dynamic model with efficiency loss significantly reduces the computation without any accuracy loss.

**Ablation Experiments on Dynamic Modeling.** We test performance of or approach by turning of dynamic modeling along temporal and channel dimensions on Mini-Kinetics-200. Table 10 shows that dynamic modeling along both dimensions obtains the best performance while requiring the least computation. This shows importance of input-dependent policy for deciding how many features need to be computed for both temporal and channel dimensions.

**Visualization and Analysis.** To better understand the policy decision process, we dissect the network layers and count the ratio of feature maps that are being computed during each convolution layers for each category. From Figure 3, we observe that: In X3D, point-wise convolutions which right after the depth-wise convolutions have more variation among classes and network tends to consume more temporal-wise features at the early stage and compute more channel-wise features at the late stage of the architecture. The channel-wise policy has also more variation than the temporal-wise policy among different categories. Furthermore, we show few contrasting examples which are in the

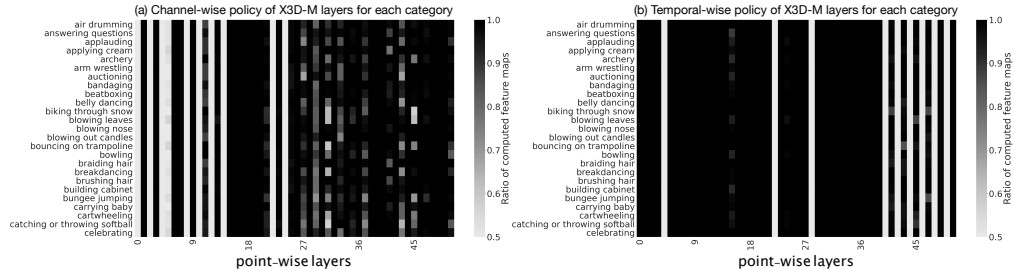

Figure 3: **Ratio of computed feature per layer and class on Mini-Kinetics-200 dataset.** We pick the first 25 classes of Mini-Kinetics-200 and visualize the per-block policy of X3D-M on each class. Lighter color means fewer feature maps are computed while darker color represents more feature maps are computed.

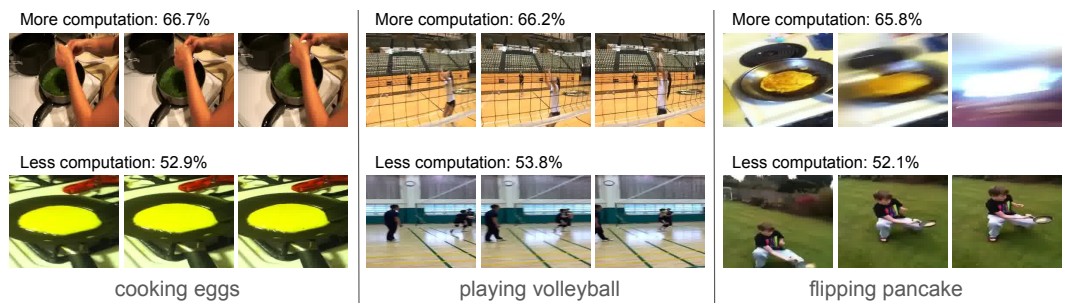

Figure 4: **Validation video clips from Mini-Kinetics-200.** For each category, we plot two input video clips which consume the most and the least computational cost respectively. We infer these video clips with 8-frame dynamic R(2+1)D-18 model trained on Mini-Kinetics-200 and the percentage indicates the ratio of actual computational cost of 2D convolution to that of the original fixed model. Best viewed in color.

Table 11: **VA-RED$^2$ on semantic segmentation.** We choose dilated ResNet-18 as our backbone architecture and set the search space as 2. Models are trained for 100K iterations with batch size of 8.

| Model | Original model | | Channel-wise reduction using VA-RED$^2$ | | | |
|---|---|---|---|---|---|---|
| | GFLOPs | mean IoU | GFLOPs$_{avg}$ | GFLOPs$_{max}$ | GFLOPs$_{min}$ | mean IoU |
| Dilated ResNet-18 | 10.6 | 31.2% | **7.8** | 9.1 | 7.3 | **31.3%** |

same category while requiring very different computation in Figure 4. Video clips which have more complicated scene configuration (e.g. cooking eggs and playing volleyball) and more violent camera motion (e.g. flipping pancake) tend to need more feature maps to do the correct predictions. More qualitative results can be found in Section E, Section F and Section G of the Appendix.

**VA-RED$^2$ on Dense Visual Tasks.** Our VA-RED$^2$ framework is also applicable for some dense visual tasks, like semantic segmentation, which requires the pixel-level prediction for the input content. To prove this, we apply our method to a semantic segmentation model on the ADE-20K dataset (Zhou et al., 2017; 2018b). We report computational cost of model encoder and the mean IoU (Intersection-Over-Union) in Table 11. As can be seen from Table 11, our proposed VA-RED$^2$ has the absolute advantage in terms of efficiency while maintaining the precision of segmentation. This experiment clearly shows that our method is not only effective on the recognition and detection tasks, but also applicable to the dense visual tasks like semantic segmentation.

## 5  CONCLUSION

In this paper, we propose an input-dependent adaptive framework called VA-RED$^2$ for efficient inference which can be easily plugged into most of existing video understanding models to significantly reduce the model computation while maintaining the accuracy. Extensive experimental results on video action recognition, spatio-temporal localization, and semantic segmentation validate the effectiveness of our framework in multiple standard benchmark datasets.

**Acknowledgements.** This work is supported by IARPA via DOI/IBC contract number D17PC00341. This work is also supported by the MIT-IBM Watson AI Lab.

**Disclaimer.** The U.S. Government is authorized to reproduce and distribute reprints for Governmental purposes notwithstanding any copyright annotation thereon. The views and conclusions contained herein are those of the authors and should not be interpreted as necessarily representing the official policies or endorsements, either expressed or implied, of IARPA, DOI/IBC, or the U.S. Government.

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

## A    DATASET DETAILS

We evaluate the performance of our approach using three video action recognition datasets, namely Mini-Kinetics-200 (Meng et al., 2020), Kinetics-400 (Carreira & Zisserman, 2017), and Moments-In-Time (Monfort et al., 2019) and one spatio-temporal action localization task namely J-HMDB-21 (Jhuang et al., 2013). Kinetics-400 is a large dataset containing 400 action classes and 240K training videos that are collected from YouTube. The Mini-Kinetics dataset contains 121K videos for training and 10K videos for testing, with each video lasting 6-10 seconds. The original Kinetics dataset is publicly available to download at `https://deepmind.com/research/open-source/kinetics`. We use the official training/validation/testing splits of Kinetics-400 and the splits released by authors in (Meng et al., 2020) for Mini-Kinetics-200 in our experiments.

Moments-in-time (Monfort et al., 2019) is a recent collection of one million labeled videos, involving actions from people, animals, objects or natural phenomena. It has 339 classes and each video clip is trimmed to 3 seconds long. This dataset is designed to have a very large set of both inter-class and intra-class variation that captures a dynamic event at different levels of abstraction (i.e. "opening" doors, curtains, mouths, even a flower opening its petals). We use the official splits in our experiments. The dataset is publicly available to download at `http://moments.csail.mit.edu/`.

Joints for the HMDB dataset (J-HMDB-21 (Jhuang et al., 2013)) is based on 928 clips from HMDB51 comprising 21 action categories. Each frame has a 2D pose annotation based on a 2D articulated human puppet model that provides scale, pose, segmentation, coarse viewpoint, and dense optical flow for the humans in action. The 21 categories are brush hair, catch, clap, climb stairs, golf, jump, kick ball, pick, pour, pull-up, push, run, shoot ball, shoot bow, shoot gun, sit, stand, swing baseball, throw, walk, wave. The dataset is available to download at `http://jhmdb.is.tue.mpg.de/`.

## B    IMPLEMENTATION DETAILS

**Details of Shared-weight Training and Inference.** In this section, we provide more details of the shared-weight mechanism presented in Section 3 of the main paper. We first compute all the possible necessary features using a big kernel and then for each dynamic convolution with different scaling factor, we sample its corresponding ratio of necessary features and reconstruct the rest features by cheap operations to get the final output. For example, the original channel-wise dynamic convolution at ratio $r = (\frac{1}{2})^{(i-1)}$ can be analogized to

$$[(f_l^c(X_l, r = (\frac{1}{2})^{i_s^c-1})[0 : (\frac{1}{2})^{(i-1)}C_{out}]), (\Phi^c(f_l^c(X_l, r = (\frac{1}{2})^{i_s^c-1})[0 : (\frac{1}{2})^{(i-1)} \cdot C_{out}]))], \quad (7)$$

where $[\cdot : \cdot]$ is the index operation along the channel dimension, and $i_s^c$ is the index of the largest channel-wise filter, during training phase, we have $i_s^c = 1$, while during inference phase, $i_s^c$ is the smallest index for $V_c^l$, $s.t. V_c^l[i_s^c] = 0$. By utilizing such a share-weight mechanism, the computation of the total channel-wise dynamic convolution is reduce to $(\frac{1}{2})^{i_s^c-1} \cdot \mathcal{C}(f_l)$. Further, we have the total computational cost of the adjunct process as

$$\mathcal{C}(f_l^{t,c}) = (\frac{1}{2})^{i_s^c+i_s^t-2} \cdot \mathcal{C}(f_l), \quad (8)$$

where $i_s^t$ is the index of largest temporal-wise filter.

**Training and Inference.** We apply our method mainly to 2D convolutions in R(2+1)D since 2D convolution takes the most computational cost compared with 1D convolution. We train most of our models on 96 NVIDIA Tesla V100-32GB GPUs and perform synchronized BN (Ioffe & Szegedy, 2015) across all the GPUs. For R(2+1)D (Tran et al., 2018), the learning rate is initialized as $0.18$ and the weight decay is set to be $5 \times 10^{-4}$. For I3D (Carreira & Zisserman, 2017; Xie et al., 2018) and X3D (Feichtenhofer, 2020), the learning rates both start from $1.8$ and weight decay factors are $1 \times 10^{-4}$ and $5 \times 10^{-5}$ respectively. Cosine learning rate decaying strategy is applied to decrease the total learning rate. All of the models are trained from scratch and warmed up for 15 epochs on mini-Kinetics/Kinetics, 8 epochs on Moments-In-Time dataset. We adopt the Nesterov momentum optimizer with an initial weight of 0.01 and a momentum of 0.9. During training, we follow the data augmentation (location jittering, horizontal flipping, corner cropping, and scale jittering) used in TSN (Wang et al., 2016) to augment the video with different sizes spatially and flip

Table 12: **Quantitative results of redundancy experiments.** We compute the correlation coefficient, RMSE and redundancy proportions (RP) for feature maps in well-known pretrained video models on Moments-in-Time and Kinetics-400 datasets. RP is calculated as the number of tensors with both CC and RMSE above redundancy thresholds of 0.85 and 0.001, respectively. We show results corresponding to averaging the per layer values for all videos in the validation sets. We observe that networks trained on Moments-In-Time (and evaluated on the Moments in Time validation set) tend to present slightly less redundancy than their Kinetics counterparts, and the time dimension tends to be more redundant than the channel dimension in all cases. We observe severe redundancy across the board (with some dataset-model pairs achieving upwards of 0.8 correlation coefficient between their feature maps), which further motivates our redundancy reduction approach.

| Dataset | Model | Dimension | CC | RMSE | RP |
|---------|-------|-----------|------|-------|------|
| Moments-In-Time | I3D | Temporal | 0.77 | 0.083 | 0.62 |
| | I3D | Channel | 0.71 | 0.112 | 0.48 |
| | R(2+1)D | Temporal | 0.73 | 0.108 | 0.49 |
| | R(2+1)D | Channel | 0.68 | 0.122 | 0.43 |
| Kinetics-400 | I3D | Temporal | 0.81 | 0.074 | 0.68 |
| | I3D | Channel | 0.76 | 0.091 | 0.61 |
| | R(2+1)D | Temporal | 0.78 | 0.081 | 0.64 |
| | R(2+1)D | Channel | 0.73 | 0.088 | 0.58 |

the video horizontally with 50% probability. We use single-clip, center-crop FLOPs as a basic unit of computational cost. Inference-time computational cost is roughly proportional to this, if a fixed number of clips and crops is used, as is for our all models. Note that Kinetics-400 dataset is shrinking in size ($\sim$15% videos removed from original Kinetics) and the original version used in (Carreira & Zisserman, 2017) are no longer available from official site, resulting in some difference of results.

## C  REDUNDANCY ANALYSIS

To motivate our redundancy reduction approach, we measure and visualize the internal redundancy of well known pretrained networks. We analyze the internal feature maps of existing pre-trained I3D-InceptionV2 and R(2+1)D networks on Moments in Time and Kinetics. For each model-dataset pair, we extract feature maps for all examples in the validation sets, in both time and channel dimensions, and measure their similarity. In detail, our method consists of the following steps: (1) For a given input, we first extract the output feature maps from all convolutional layers in the network at hand. (2) In each layer, we measure the similarity of each feature map to each other with Person's correlation coefficient (CC) and root mean squared error (RMSE). We additionally flag feature maps that exhibit high similarity as redundant. (3) After computing this for the validation sets, we average the values over all examples to obtain mean metrics of redundancy per model and per dataset. We additionally compute the ranges of these values to visualize how much redundancy can vary in a model-dataset pair. We present quantitative results in Table 12 and show examples of our findings in Figure 5.

## D  VA-RED$^2$ ON LONGER-TRAINING MODEL

In our experiments, all of our models are trained under a common evaluation protocol for a fair comparison. To balance the training cost and model performance, we use a smaller epoch size than the original paper to train our models. For example, authors in (Tran et al., 2018) and (Feichtenhofer, 2020), train the R(2+1)D models and X3D models for 188 epochs and 256 epochs respectively to pursue the state-of-the art. However, we only train the models for 120 epochs to largely save the computation resources and training time. However, to rule out the possibility that our base models (i.e., without using Dynamic Convolution) benefit from longer training epochs while our VA-RED$^2$ may not, we conduct an ablation study on the epoch size in Table 13. We can see that our method still shows superiority over the base model in terms of the computational cost and accuracy on the 256-epoch model. Thus we conclude that the effectiveness of our method in achieving higher performance with low computation also holds on the longer-training models.

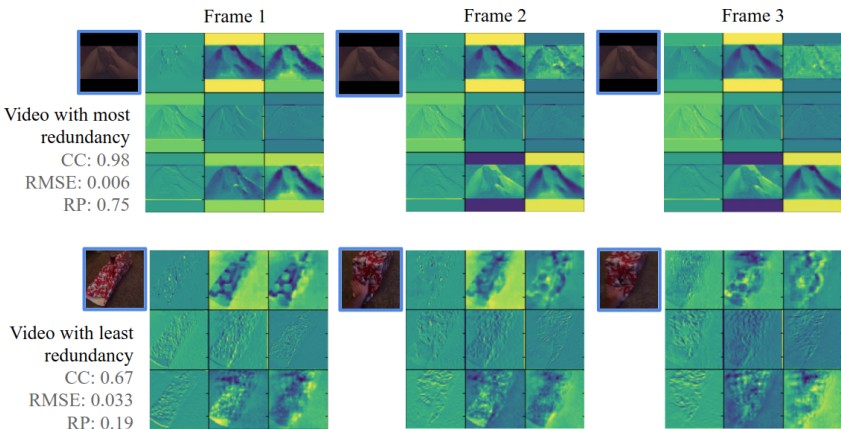

Figure 5: Visualization of the first 9 filters of the first layer of I3D, on examples with most (top) and least (bottom) redundancy in the temporal dimension. We exemplify the results on frames 1, 2 and 3. As can be seen, the video with most redundancy consists of a relatively static video with little movement, and the sets of feature maps from frame to frame harbor heavy similarity. The video with least redundancy consists of a gift unwrapping with rapid movement (even in the first few frames) and the corresponding feature maps present visible structural differences from frame to frame. Although in both cases, redundancy is present, it is clear that some examples present much more redundancy than others, thus motivating our input-dependent redundancy reduction approach.

Table 13: **Comparison between the performance of VA-RED$^2$ on 120-epoch X3D model and 256-epoch X3D model.** We choose X3D-M as our backbone architecture and set the search space as 2. We train one group of models for 120 epochs and the other for 256 epochs.

| Model | Dynamic | 120 epochs | | | | 256 epochs | | | |
|---|---|---|---|---|---|---|---|---|---|
| | | GFLOPs | clip-1 | video-1 | video-5 | GFLOPs | clip-1 | video-1 | video-5 |
| X3D-M | ✗ | 6.42 | 63.2 | 70.6 | 90.0 | 6.42 | 64.4 | 72.3 | 90.8 |
| | ✓ | 5.38 | **65.3** | **72.4** | **90.7** | 5.87 | **66.4** | **73.6** | **91.2** |

## E  FEATURE MAP VISUALIZATIONS

To further validate our initial motivation, we visualize the feature maps which are fully computed by the original convlution operation and those which are generated by the cheap operations. We demonstrate those in both temporal dimension (c.f. Figure 6) and channel dimension (c.f. Figure 7). In both cases we can see that the proposed cheap operation generates meaningful feature maps and some of them looks even no difference from the original feature maps.

## F  POLICY VISUALIZATIONS

To compare with the policy on Mini-Kinetics-200 (Figure 3 of the main paper), we also visualize the ratio of features which are consumed in each layer on Kinetics-400 (c.f. Figure 8) and Moments-In-Time (c.f. Figure 9). We can see from these two figures that the conclusions we draw from Mini-Kientics-200 still hold. Specifically, In X3D, point-wise convolutions which right after the depth-wise convolutions have more variation among classes and network tends to consume more temporal-wise features at the early stage and compute more channel-wise features at the late stage of the architecture. However, R(2+1)D choose to select fewer features at early stage by both temporal-wise and channel-wise policy. Furthermore, we count the FLOPs of each instance on Mini-Kinetics-200, Kinetics-400, and Moments-In-Time and plot pie charts to visualize the the distribution of this instance-level computational cost. We analyze such distribution with two models: R(2+1)D-18 and X3D-M. All of the results are demonstrated in Figure 10.

# G  QUALITATIVE RESULTS

We show additional input examples which consume different levels of computational cost on Kinetics-400 dataset (c.f. Figure 11) and Moments-In-Time dataset (c.f. Figure 12). To be consistent, we

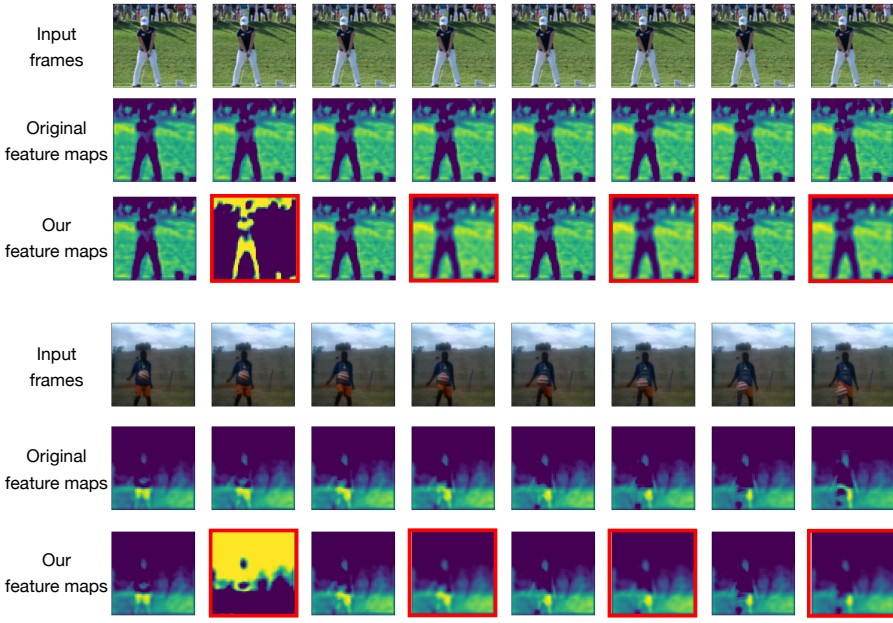

Figure 6: **Visualization of temporal-wise feature maps.** We plot the temporal feature maps which are fully computed by the original convolution and those mixed with cheaply generated feature maps. The feature maps marked with red bounding boxes are cheaply generated. We do this analysis on 8-frame dynamic R(2+1)D-18 pretrained on Mini-Kinetics-200. These feature maps are the output of the first spatial convolution combined with ReLU non-linearity inside the `ResBlock_1`. We can see that most of the cheaply generated feature maps looks no difference from the original feature maps, which further support our approach. Best viewed in color.

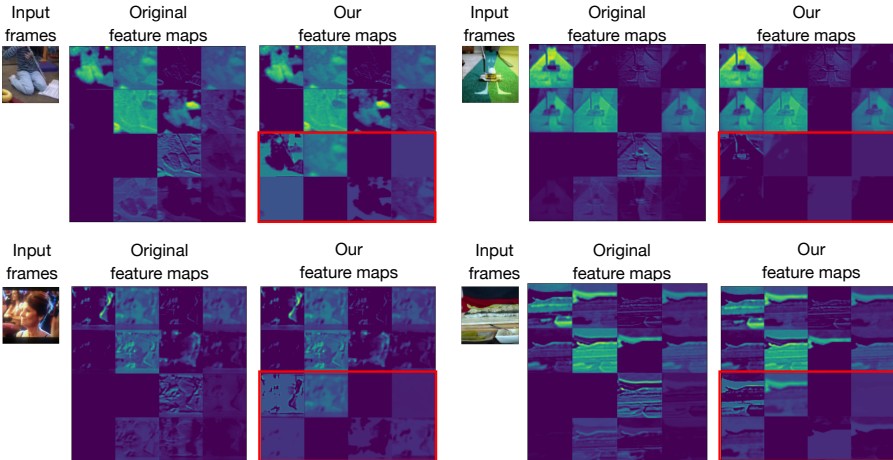

Figure 7: **Visualization of channel-wise feature maps.** We plot the feature maps across the channel dimension. We contrast two kinds of feature maps: fully computed by the original convolution and those mixed with cheaply generated feature maps. The feature maps inside the red bounding boxes are cheaply generated. The analysis is performed on 8-frame dynamic R(2+1)D-18 model which is pretrained on Mini-Kinetics-200 dataset and we extract these feature maps which are output by the first spatial convolution layer inside the `ResBlock_1`. Best viewed in color.

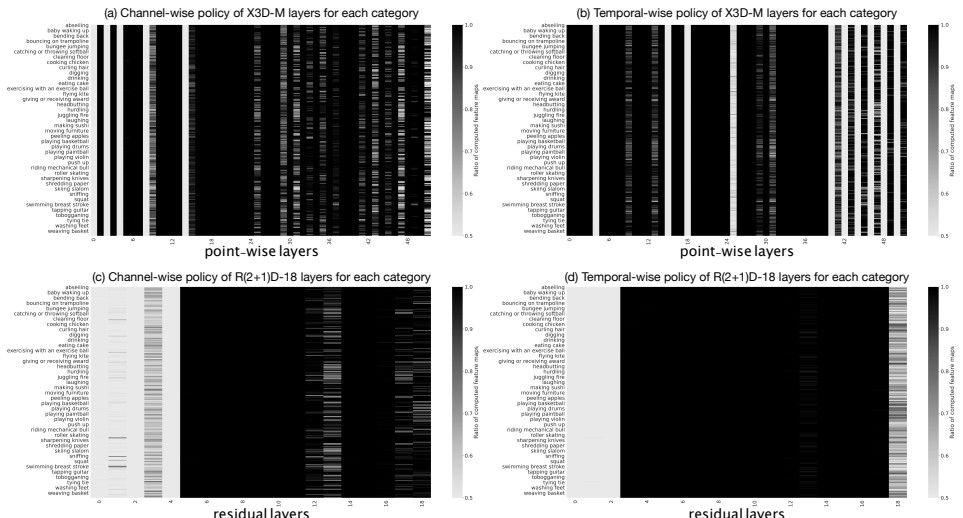

Figure 8: **Ratio of computed feature per layer and class on Kinetics-400 dataset.** We visualize the per-block policy of X3D-M and R(2+1)D-18 on all 400 classes. Lighter color means fewer feature maps are computed while darker color represents more feature maps are computed. While X3D-M tends to consume more temporal-wise features at the early stage and compute more channel-wise features at the late stage, R(2+1)D choose to select fewer features at early stage by both temporal-wise and channel-wise policy. For both architectures, the channel-wise policy has more variation than the temporal-wise policy among different categories.

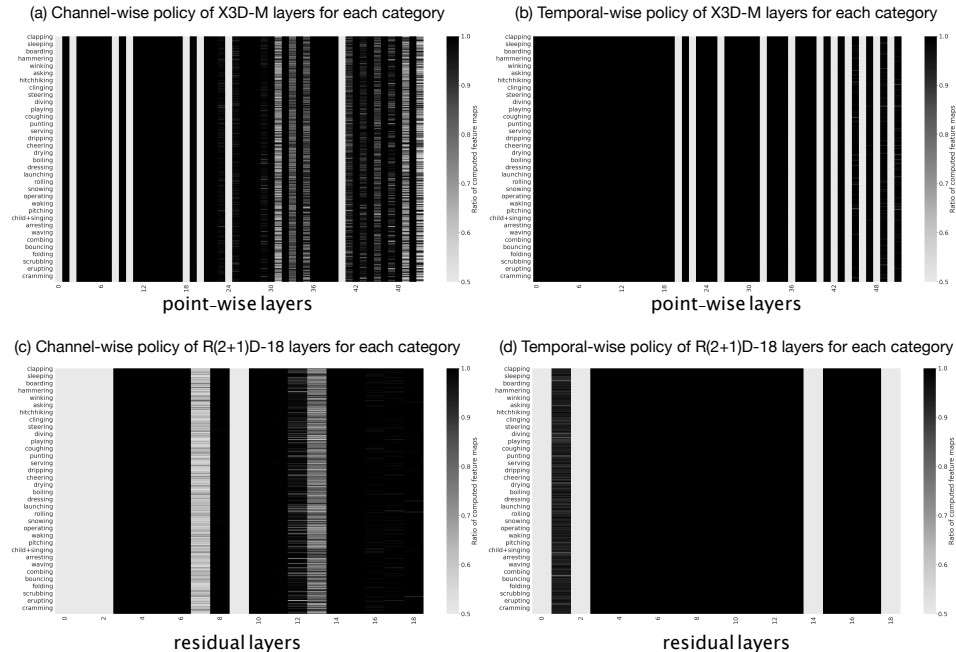

Figure 9: **Ratio of computed feature per layer and class on Moments-In-Time dataset.** We visualize the per-block policy of X3D-M and R(2+1)D-18 on all 339 classes. Lighter color means fewer feature maps are computed while darker color represents more feature maps are computed.

use the 16-frame dynamic R(2+1)D-18 as our pre-trained model. We can see that the examples consuming less computation tend to have less temporal motion, like the second example in Figure 11, or have a relatively simple scene configuration, like the first and second examples in Figure 12.

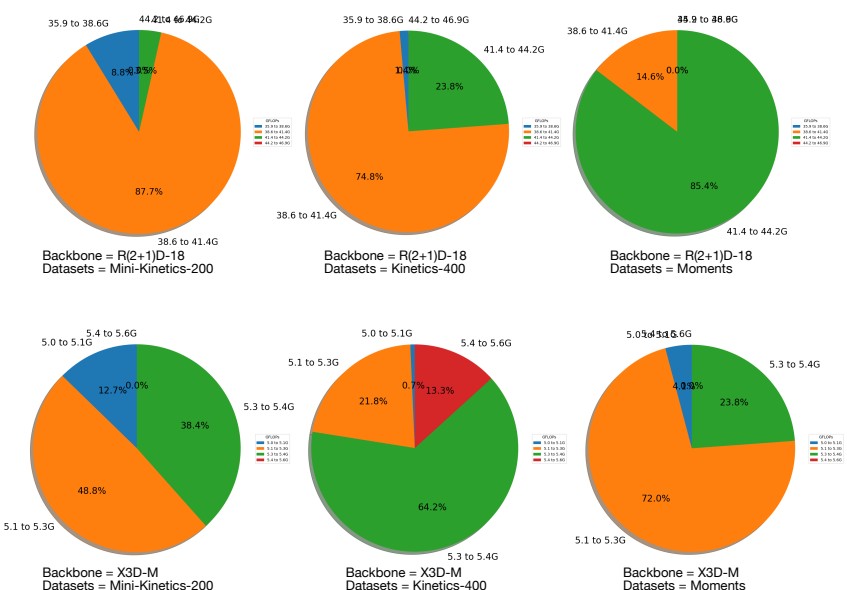

Figure 10: **Computational cost distribution across different models on different datasets.** We count the computation of each instance cost by different models on different datasets. For instance, for the upper-left one, we use the model backbone of R(2+1)D-18 on Mini-Kinetics-200. This sub-figure indicates that there are $87.7\%$ of videos in Mini-Kinetics-200 (Dataset) consuming $38.6 - 41.4$ GFLOPs by using R(2+1)D-18 (Backbone), $8.8\%$ of videos consuming $35.9 - 38.6$ GFLOPs, and $3.5\%$ of videos consuming $41.4 - 44.2$ GFLOPs.

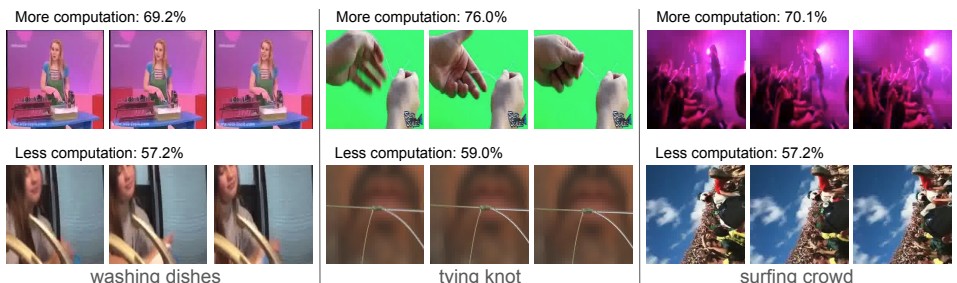

Figure 11: **Validation video clips from Kinetics-400.** For each category, we plot two input video clips which consume the most and the least computational cost respectively. We infer these video clips with 16-frame dynamic R(2+1)D-18 which is pre-trained on Kinetics-400. The percentage in the figure indicates the ratio of the actual computational cost of 2D convolution to that of the original fixed model. Best viewed in color.

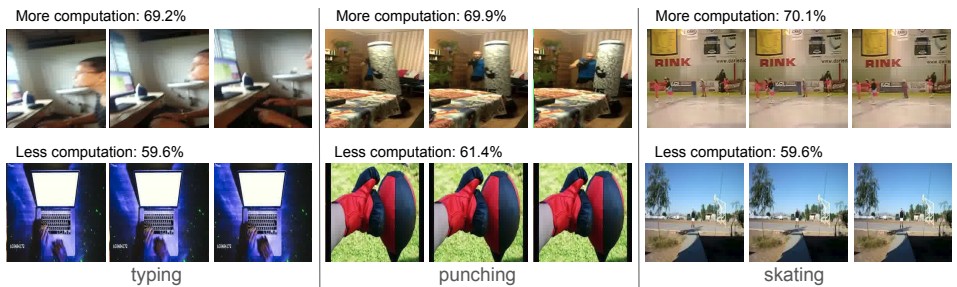

Figure 12: **Validation video clips from Moments-In-Time.** For each category, we plot two input video clips which consume the most and the least computational cost respectively. We infer these video clips with 16-frame dynamic R(2+1)D-18 which is pre-trained on Moments-In-Time. The percentage in the figure indicates the ratio of the actual computational cost of 2D convolution to that of the original fixed model. Best viewed in color.

