# OpenReview forum: "VA-RED$^2$: Video Adaptive Redundancy Reduction"
_ICLR.cc/2021/Conference — ICLR 2021 Poster_

### Official Review · AnonReviewer1 · 2020-10-18
**This paper proposes a novel framework called $\text{VA-RED}^2$ to reduce spatial and temporal features to be computed for video understanding, which can reduce FLOPs when inferencing the video but remains the performance.  This paper is well-written and conducts extensive experiments to validate the performance of proposed approach.**

**Rating:** 6
**Confidence:** 1

**Review:**

This paper proposes a novel framework called $\text{VA-RED}^2$ to reduce spatial and temporal features to be computed for video understanding, which can reduce FLOPs when inferencing the video but remains the performance.

The authors have done extensive experiments on video action recognition tasks and spatio-temporal action localization task in the area of video understanding. For the video action recognition task, experiments are carried out using Mini-Kinetics-200, Kinetics-400, and Moments-In-time datasets. For the action localization task, J-HMDB-21 dataset is used. Results show that this framework is promising, which reduces the computation but main the performance.


Question:
1. X3D-M in the original paper achieved at top-1 74.6 for Kinetics-400 dataset (but in table 4 reports clip-1: 61.8, video-1: 67.9) and FLOPs is 4.73 (6.20 reported in the paper). Can authors explain why there is a difference here?


Minor:
1. Reduce ratio fact for channel-wise dynamic convolution $r=\frac{1}{2}^{p_c}$ such as in Fig.2 on page 4 and equation 7, equation 8 in supplementary materials on page 13. I think it would make more sense representing it as $(\frac{1}{2})^{p_c}$.

---

> ### Author Response · Authors · 2020-11-23
> **Response to AnonReviewer1**
>
> We appreciate R1 for giving us such positive comments on the advantages of our paper! Below we address R1’s concern on results of X3D model on Kinetics-400.
>
> **(a) Difference in X3D-M results:** Until the submission of this paper, the X3D authors hadn't released their official code for the implementation of their models. Thus, despite our best efforts, the implementation details of our reproduced version may be slightly different from the original X3D. During the rebuttal period, we re-launched our  X3D experiments on Kinetics-400 with the official implementation which was released recently and updated the results in Table 5 of the revised version. Our adaptive approach still outperforms X3D by 1.8% in top-1 video accuracy, while reducing computation (GFLOPs) by a margin of about 16.1%  on Kinetics-400 dataset. Note that while the updated performance of X3D is improved by 2.7% (67.9 vs 70.6),  it is still different from the original paper, which we believe is due to the difference in dataset size as Kinetics-400 is shrinking in size (about ~15% of the videos are missing) and the original version is no longer available from the official site, resulting in some difference of results. Another possible reason could be due to some additional tricks used by X3D to further boost the performance that are currently not included in the official implementation. The main difference in FLOPs comes from the fact that we report our results based on 256x256 input resolution while they show results with 224x224 input resolution.
>
> **(b) Reduce ratio fact for channel-wise dynamic convolution.** Thanks for R1’s kindly reminder, we have revised this in our updated PDF.

---

### Official Review · AnonReviewer2 · 2020-10-25
**This submission proposes to adaptively reduce redundancy in temporal and channel dimension to make existing video networks more efficient.**

**Rating:** 6
**Confidence:** 4

**Review:**

Pros:
1. The motivation of the submission is clear and the writing is easy to follow. The problem being addressed is a practical and important issue in deploying video models to real applications.

2. Experiments are thorough. This submission has done experiments on both action recognition and action localization task. The performance is promising on most video networks.

Cons:
1. This submission proposes to address the problem of slow video inference, but the only metric they report is the Gflops. We know that Gflops is not a good indicator for speed comparison. Sometimes, a small Gflops does not mean fast speed. So I recommend to add a "speed" column in every table of this submission, to compare the actual inference speed among the methods.

2. Regarding the searching method, can authors justify what are the differences to X3D? X3D also search the optimal combination of temporal dimension and channel dimension. Are the differences only on adding efficiency loss?

3. The efficiency loss is widely used in searching efficient image models, like in Proxylessnas. What is the contribution and motivation of using it here in searching video models?

---

> ### Author Response · Authors · 2020-11-23
> **Response to AnonReviewer2**
>
> We would like to thank R2 for acknowledging that our paper is addressing a practical and important problem, and that the results of our paper are promising!
>
> **(a) Speed comparison.** We appreciate R2 for this constructive suggestion on speed. The GFLOPs indeed can not completely reflect the actual speed of each model, particularly on GPUs.  Following R2’s suggestion, we add the “speed”  (measured in terms of clip/second, which denotes the number of video clips that are processed in one second) column to most of our tables in the revised version. We created the environment with PyTorch 1.6, CUDA 11.0, and a single NVIDIA TITAN RTX (24GB) GPU as our testbed. Although we devise our framework mainly based on the efficiency in terms of FLOPs, our method still has advantages regarding actual speed. For instance, for the I3D_v2 model with 16 frames on the action recognition task, our method boosts its speed by 21.7% (141.7 clip/second vs 116.4 clip/second). Likewise, for the R(2+1)D  model with 16 frames, our method boosts its speed by 10.6% (108.7 clip/second vs 97.1 clip/second) (Table 2 in the revised version).
>
> **(b) Differences in search w.r.t X3D.** Our search strategy is completely different from X3D in three different aspects. **(a) Objective of searching.** What X3D searches for is a fixed optimal combination of temporal stride, channel width, input resolution and model depth. On the other hand, we search for a policy which dynamically predicts the ratio of features which are redundant for each input instance so that we can replace them with cheaply-obtained features. In other words, our proposed approach makes decisions on a per-input basis, leading to different amounts of computation for different videos,  in contrast to the one-size-fits-all scheme used by X3D.  **(b) Searching methodology.** X3D follows a heuristic searching methodology, which gradually expands the dimension of each factor and manually compares models with different expanded factors. However, our method utilizes a learning-based searching methodology, which adopts light-weight policy layers with the efficiency loss to learn to decide how many features are needed. **(c) Searching direction.** X3D starts searching from an efficient simple base model with gradually increasing FLOPs. However, our method starts from a redundant base model and searches to reduce the FLOPs. Extensive experiments on multiple benchmark datasets show that our dynamic approach achieves 16% reduction in computation over X3D without any performance loss, for video action recognition tasks. Besides X3D, our approach also outperforms R2+1D, I3D, TPN, CorrNet in both accuracy and efficiency,  which shows that our approach is model-agnostic, and hence can be served as a plugin operation for a wide range of action recognition architectures.
>
> **(c) Motivation and contribution of efficiency loss.** The goal of our paper is to customize the dynamic network architecture with the minimal computation for each input instance. Specifically, the core idea of this paper is that we want to dynamically decide how many features should be replaced by the cheaply-obtained feature. Thus the motivation of devising our efficiency loss is to maximize the ratio of cheaply-obtained features during the training phase, which is a necessary part of the whole VA-RED^2 framework. And the contribution of the efficiency loss is that it significantly reduces the computational cost as  can be  seen from Table 9 of the main paper.

---

### Official Review · AnonReviewer3 · 2020-10-28
**Review for VA-RED: Video Adaptive Redundancy Reduction**

**Rating:** 6
**Confidence:** 4

**Review:**

Summary
The paper presents a framework to reduce internal redundancy in the video recognition model. To do so, given the input frames, the framework predicts two scaling factors to conduct temporal and channel dimension reduction. The remaining part is reconstructed by cheap operations. The authors show that the framework achieves favorable results on several benchmarks.

Strengths
+ The paper is well written.
+ Strong quantitative and qualitative results (Consistent improvement over the state-of-the-art baselines).
+ Solid ablation studies and analysis.

Weakness
- The core idea of using a light-weight neural module to maximize the video model efficiency is not novel.
For example, AR-Net already has shown that a policy network can decide the video input resolution, i.e., spatial dimension, adaptively, leading to improve both efficiency and accuracy. In this work, the key concept is basically the same and only the primitive operation unit, i.e., temporal and channel dimensions, is different.

- Please provide experiment-level comparisons with AR-Net

- The following SOTA video models are missing in the reference.
DynamoNet: Dynamic action and motion network, ICCV 2019
Video Modeling with Correlation Networks, CVPR 2020
RubiksNet: Learnable 3D-Shift for Efficient Video Action Recognition, ECCV2020
Directional Temporal Modeling for Action Recognition, ECCV2020

Question

- Can the idea be applied to dense video understanding tasks, such as video semantic segmentation?

Rating

- Aiming at acquiring an efficient model in a data-driven manner is indeed important for video models.
While the key idea is not novel, the paper has obvious empirical contributions that may help communities to invigorate future researches in this direction.

---

> ### Author Response · Authors · 2020-11-23
> **Response to AnonReviewer3**
>
> We would like to thank R3 for confirming our results are promising and the paper is well written!
>
> **(a) Differences with AR-Net (Meng et al. 2020).** The key differences between our method and AR-Net lies at: **(a) Insight of reducing FLOPs.** AR-Net accelerates the video inference by adaptively choosing the minimal input resolution, i.e. spatial width to correctly predict the results without considering the relationship between features from consecutive frames. On the other hand, our method focuses on reducing the temporal redundancy among the features of a series of consecutive frames and the spatial redundancy among a set of output features by replacing the redundant features with cheaply-obtained features. Thus, our method and AR-Net are fundamentally quite different as AR-Net does not reconstruct redundant feature maps using cheap linear operations. **(b) Model memory.** AR-Net adopts several independent networks corresponding to different input resolutions to adaptively infer the video, which makes the model size significantly larger than the original model. While our method integrates all of our possible network architectures into a shared-weight super network, the parameter size of whom is just slightly larger (almost the same) than the original model. **(c) Scope of application.** AR-Net is only applicable to 2D video models, like Temporal Segment Network (TSN). In contrast,  our VA-RED^2 framework is model-agnostic and hence can be applied to a wide- variety of action recognition architectures including  2D, (2+1)D, and 3D video models, like (R(2+1)D, X3D, I3D). Moreover, while AR-Net is only limited to action recognition, our dynamic framework is generalizable to not only video action recognition but also to spatio-temporal action localization and dense segmentation tasks, achieving promising results over competing methods.
>
> **(b) Experimental-level comparison with AR-Net.** Following the reviewer’s suggestion, we compare our method with AR-Net using 16-frame TSN ResNet50 on Mini-Kinetics-200 and observe that our method has advantages on the FLOPs, model size, and performance,  e.g., our approach outperforms AR-Net in both top-1 clip accuracy  (67.2 vs 68.3) and FLOPS (44.8G vs 43.4G), while using about 62% less parameters (63.0M vs 23.9M). We have added this result in the updated version ( see Table 4 (Right) of the revised paper).
>
> **(c) Missing references.** Thanks for pointing out these to us! We have updated the related work section to reflect these relevant works in the revised version. Besides, we conducted additional experiments of our method with one of these SOTA models, CorrNet [1] with R(2+1)D backbone, the results of which are listed in Table 4 (Left) of the revised paper. Experimental results show that our dynamic approach outperforms the baseline CorrNet by 1.8% in top-1 video accuracy (68.2 vs 70.0), while reducing the computational cost/FLOPS  by 25.2%  (60.8G vs 45.5G) on Mini-Kinetics-200 dataset.
>
> **(d) VA-RED^2 with dense video understanding tasks.** Our adaptive approach for redundancy reduction is a general framework which can be applied to different types of video  tasks, not limited to action recognition. Specifically, our method can be applied to dense video understanding tasks, like video semantic understanding, as long as there exists redundancy in the temporal or channel dimension of videos. To prove this, we conducted an experiment by applying our method to a semantic segmentation model to remove the channel-wise redundancy. The results are reported in Table 11 in our updated paper. The experiments are conducted on ADE20K dataset [2] with the PSPNet [3], where we choose the dilated ResNet-18 as our backbone architecture. We can see that our method significantly boosts the efficiency of the segmentation model as our method can save around 26.4% computational cost of the original model while maintaining the accuracy.
>
> References:
> [1] Wang et al. Video Modeling with Correlation Networks, CVPR 2020.
> [2] Zhou et al. Scene Parsing through ADE20K Dataset, CVPR 2017.
> [3] Zhao et al. Pyramid Scene Parsing Network. CVPR 2017.

---

### Author Response · Authors · 2020-11-23
**Summary of Changes and Paper Revision**

We would like to thank all the reviewers for their constructive comments! We are  glad that the reviewers found that: (a) our paper proposes a novel framework (R3) for video redundancy reduction, with clear motivation (R2); (b) our work addresses a practical and important issue of deploying video models to real applications (R2); (c) our experiments are through with solid ablation studies and qualitative analysis, showing promising performance (consistent improvement over state-of-the-art baselines) on both video action recognition tasks and spatio-temporal action localization task in the area of video understanding (R1, R2, R3).

We have addressed all the questions that the reviewers posed with additional experimental comparisons  and clarifications. All of these additional experiments and suggestions have been added into the updated PDF (changes are highlighted in blue). Below,  we summarize the main changes to the paper and encourage the reviewers to take a look at the new additions.

1. Updated X3D results on Kinetics-400,
2. Added speed comparison in the results,
3. Added comparison with Corrnet (Wang et al. 2020),
4. Added comparison with AR-Net (Meng et al. 2020),
5. Added results on dense  semantic segmentation task,
6. Updated related work to reflect all the relevant works.

---

### Decision · Program_Chairs · 2021-01-07
**Final Decision**

**Decision:**

Accept (Poster)

**Comment:**

This paper presents work on efficient video analysis.  The reviewers appreciated the clear formulation and effective methodology.  Concerns were raised over empirical validation.  The authors' responses added additional material that assisted in clarifying these points.  After the discussion the reviewers converged on a unanimous accept rating.  The paper contains solid advances in efficient inference for video analysis and is ready for publication.